# Observation of a thermoelectric Hall plateau in the extreme quantum limit

Wenjie Zhang[1,9], Peipei Wang[2,9], Brian Skinner [3,4,9], Ran Bi[1], Vladyslav Kozii [3,5,6], Chang-Woo Cho[2], Ruidan Zhong[7], John Schneeloch[7], Dapeng Yu [2], Genda Gu[7], Liang Fu[3✉], Xiaosong Wu [1,8✉] & Liyuan Zhang [2✉]

The thermoelectric Hall effect is the generation of a transverse heat current upon applying an electric field in the presence of a magnetic field. Here, we demonstrate that the thermoelectric Hall conductivity $\alpha_{xy}$ in the three-dimensional Dirac semimetal ZrTe$_5$ acquires a robust plateau in the extreme quantum limit of magnetic field. The plateau value is independent of the field strength, disorder strength, carrier concentration, or carrier sign. We explain this plateau theoretically and show that it is a unique signature of three-dimensional Dirac or Weyl electrons in the extreme quantum limit. We further find that other thermoelectric coefficients, such as the thermopower and Nernst coefficient, are greatly enhanced over their zero-field values even at relatively low fields.

[1] State Key Laboratory for Artificial Microstructure and Mesoscopic Physics, Beijing Key Laboratory of Quantum Devices, Peking University, 100871 Beijing, China. [2] Department of Physics, Southern University of Science and Technology, 518055 Shenzhen, China. [3] Department of Physics, Massachusetts Institute of Technology, Cambridge, MA 02139, USA. [4] Department of Physics, Ohio State University, Columbus, OH 43210, USA. [5] Department of Physics, University of California, Berkeley, CA 94720, USA. [6] Materials Sciences Division, Lawrence Berkeley National Laboratory, Berkeley, CA 94720, USA. [7] Condensed Matter Physics and Materials Science Department, Brookhaven National Laboratory, Upton, NY 11973, USA. [8] Frontiers Science Center for Nano-optoelectronics and Collaborative Innovation Center of Quantum Matter, Peking University, 100871 Beijing, China. [9] These authors contributed equally: Wenjie Zhang, Peipei Wang, Brian Skinner. ✉email: liangfu@mit.edu; xswu@pku.edu.cn; zhangly@sustech.edu.cn

D irac materials offer the promise of uncommonly robust transport properties. For example, two-dimensional graphene exhibits an optical absorption that is a universal constant over a wide range of frequencies[1–3], and three-dimensional (3D) Dirac materials can display enormous electrical mobility[4] and photogalvanic response[5,6]. These capabilities of Dirac materials arise from their occupying a classification intermediate between the conventional dichotomy of metals and insulators: like insulators, Dirac materials have vanishing carrier density and density of states when not doped, and, like metals, they have no energy gap to electrical and thermal excitations.

This combination of properties becomes especially tantalizing when applied to the thermoelectric effect, which is the generation of electrical voltage from a temperature gradient. A recent theoretical study suggested that the thermoelectric Hall conductivity in three-dimensional Dirac or Weyl semimetals acquires a universal plateau value at sufficiently large magnetic fields[7]. This value is independent of disorder, carrier concentration, or magnetic field, and leads to the unbounded growth of the thermopower and the thermoelectric figure of merit with increasing field[8]. These findings imply a potential pathway for achieving efficient platforms for waste heat recovery or solid state refrigeration[9].

In this work we report on the observation of a robust plateau in the thermoelectric Hall conductivity $\alpha_{xy}$ in strong magnetic fields in the Dirac semimetal ZrTe$_5$. The measured value of $\alpha_{xy}$ agrees well with the theoretical predictions. Furthermore, we observe a highly sensitive field dependence of the thermoelectric responses—including $\alpha_{xy}$, the thermopower $S_{xx}$, and the Nernst coefficient

$S_{xy}$—at $T \approx 90$ K, leading to enormous enhancement of the thermoelectric properties at relatively low fields $\leq 1$ T, which can even be achieved by permanent magnets.

## Results

**Electron bands in ZrTe$_5$.** Our observation of the plateau in the thermoelectric Hall conductivity and the rapid growth of the thermopower in ZrTe$_5$ is enabled by two key factors. First, the material is metallic with an ultralow electron concentration, which allows the system to reach the extreme quantum limit (EQL) of magnetic field already at $B \approx 1$ T. Second, the high mobility of our samples, $\mu \approx 640,000$ cm$^2$/Vs, implies that the system reaches the dissipationless limit for electron transport already at $B \approx 0.1$ T. Taken together, these factors allow us to observe the robust thermoelectric Hall response in the EQL, as well as significant enhancement of thermoelectric properties at low fields.

In terms of its electronic properties, bulk ZrTe$_5$ lies somewhere between a strong topological insulator and a weak one, depending sensitively on the crystal lattice constant[10]. At this phase boundary the band structure realizes a gapless Dirac dispersion around the $\Gamma$ point. Because of the sensitivity of the band structure, indications of all three phases have been reported by various groups[11–17]. There seems to be a consensus that the band structure depends on the growth method, defect concentration, or thickness of the sample[18–21]. Our 3D bulk samples, grown by the tellurium flux method at Brookhaven National Laboratory, have been consistently found to be 3D Dirac semimetals with no gap, or at most a very small gap below detection limits, in multiple

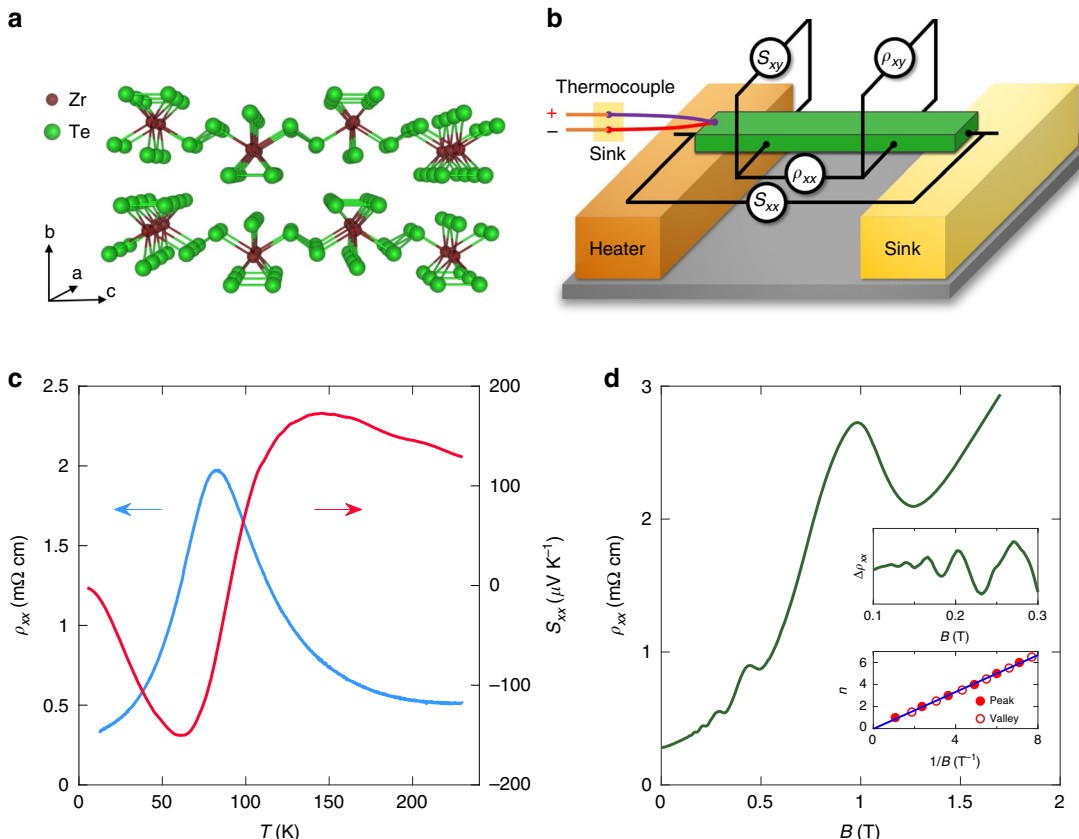

**Fig. 1 Measurement setup and resistivity. a** The crystal structure of ZrTe$_5$. **b** The measurement setup for resistivity and thermoelectric measurements. **c** Temperature dependence of the electrical resistivity (blue curve, left vertical axis) and the thermopower (red curve, right vertical axis). **d** Magnetoresistance at 1.5 K. On top of a strong positive magnetoresistance, SdH oscillations are evident (upper inset). The onset field of the oscillations is 0.13 T, indicating a high mobility. The system enters the EQL at $\approx 2$ T. The lower inset shows the index $n$ of the minima and maxima of each oscillation as a function of $1/B$.

studies by different techniques, such as ARPES, magneto-optical spectroscopy and transport[11–15,17,22]. Interestingly, a gap of 10 meV develops when the thickness of the sample is reduced to 180 nm, further confirming the sensitivity[18]. Since $ZrTe_5$ grown under certain conditions possesses a single Dirac cone and can be grown with very low carrier concentration and very high mobility, it represents an ideal platform for studying fundamental properties and possible applications of 3D Dirac electrons both in weak and strong magnetic fields. The possibility of a small band gap, and its influence on the thermoelectric Hall conductivity, are discussed in more detail in Supplementary Note 6.

**Temperature and magnetic field dependence of resistivity**. The electron concentration in our samples, as measured by the Hall effect, is $n_{Hall} \approx 5 \times 10^{16}\,cm^{-3}$ at low temperature. As we show below, at such low densities the EQL is reached already at fields $\gtrsim 2\,T$. The Hall effect remains linear in the magnetic field $B$ up to 6 T at low temperatures, which is much higher than the quantum limit field. This linearity reflects a single band of carriers, indicating that we have a simple Dirac system. As in previous studies of $ZrTe_5$[17,23], the Hall concentration $n_{Hall}$ is seen to evolve with both temperature and magnetic field, with the sample changing from $n$-type to $p$-type as $T$ is increased above $\approx 83\,K$. Previous studies suggest that this change results from a temperature-dependent Lifshitz transition[22,23]; a more thorough discussion of the electron and hole concentrations as a function of temperature is presented in Supplementary Note 2. Since the sign of the thermopower $S_{xx}$ is determined by the carrier type, i.e., positive for holes and negative for electrons, the shift from $n$-type to $p$-type transport with increasing temperature is also reflected in the temperature dependence of $S_{xx}$, which reverses its sign at $T \approx 90\,K$, as shown in Fig. 1c[24]. At the critical temperature where the carrier density is the lowest, the resistivity is at its maximum.

As the magnetic field is increased from zero, the resistivity undergoes Shubnikov-de Haas (SdH) oscillations associated with depopulation of high Landau levels. These oscillations are plotted in Fig. 1d, which shows that the EQL is achieved at all fields > 2 T. The appearance of SdH oscillations at very low field ($\approx 0.1\,T$) reflects the high mobility of our samples, $\mu \approx 640,000\,cm^2\,V^{-1}\,s^{-1}$. A previous study reported measurements of the SdH oscillations for the magnetic field oriented along the $x$, $y$, and $z$ axes, from which the Fermi surface morphology is obtained[17]. The Fermi surface is an ellipsoid with the longest principal axis in the $z$ direction. The carrier density estimated from SdH oscillations is in good agreement with $n_{Hall}$, confirming the dominance of a single band at the Fermi level at low temperature. The corresponding Fermi level is only 11 meV above the Dirac point at $T = 1.5\,K$.

**Thermoelectric coefficients in the extreme quantum limit**. Because of the low carrier density, the system enters the lowest ($N = 0$) Landau level at $B > 2\,T$ at low temperature. As the temperature is increased, the Fermi level shifts towards the Dirac point, implying that the quantum limit is reached at an even lower field. Therefore, the system is well within the EQL for a large range of magnetic field, which we sweep up to 14 T.

Our measurements of the longitudinal (Seebeck, $S_{xx}$) and transverse (Nernst, $S_{xy}$) thermoelectric coefficients are shown in Fig. 2 as a function of magnetic field. There is a general increase in the magnitude of both $S_{xx}$ and $S_{xy}$ with magnetic field in the EQL. Indeed, at $T \approx 90\,K$, where the carrier density is the lowest, $S_{xx}$ becomes as large as 800 $\mu V/K$, while $S_{xy}$ becomes larger than 1200 $\mu V/K$. The theoretical interpretation of $S_{xx}$ and $S_{xy}$, however, is complicated by the variation of the carrier density with $T$ and $B$. Indeed, the change in sign of $S_{xx}$ with $B$ at higher temperatures is likely related to the proximity of the system to a transition from

$n$-type to $p$-type conduction (as mentioned above), as is the sharp variation in $S_{xx}$ with $B$ at low fields and $T \approx 90\,K$. This variation of the carrier concentration with $B$ seems to blunt the large, linear enhancement of $S_{xx}$ with $B$ predicted in ref. [8] for higher temperatures. (A similar shift in chemical potential with B field may be responsible for the nonmonotonic dependence of $S_{xx}$ on $B$ seen in an organic Dirac material[25].)

These complications lead us to examine a more fundamental quantity, the thermoelectric conductivity $\hat{\alpha} = \hat{\rho}^{-1}\hat{S}$. Here $\hat{\rho}$ and $\hat{S}$ denote the resistivity and thermoelectric tensors, respectively, so that both the longitudinal and transverse components of the tensor $\hat{\alpha}$ can be deduced from our measurements. We focus, in particular, on the thermoelectric Hall conductivity $\alpha_{xy}$, which is plotted in Fig. 3 for the EQL. While $\alpha_{xy}$ depends in general on both $T$ and $B$, Fig. 3b shows that deep in the EQL $\alpha_{xy}$ achieves a plateau that is independent of magnetic field. Furthermore, this plateau value of $\alpha_{xy}$ is linear in temperature at temperatures $T \lesssim 100\,K$, which suggests that $\alpha_{xy} / T$ is a constant in the EQL, independent of $B$ or $T$ (Fig. 3a, see also Supplementary Note 3).

## Discussion

This strikingly universal value of $\alpha_{xy} / T$ can be understood using the following argument. The thermoelectric Hall conductivity can be defined by $\alpha_{xy} = J_y^Q/(TE_x)$, where $J_y^Q$ is the heat current density in the $y$ direction under conditions where an electric field $E_x$ is applied in the $x$ direction and the temperature $T$ is uniform. In the limit of large magnetic field (large Hall angle), electrons drift perpendicular to the electric field via the $\vec{E} \times \vec{B}$ drift, and thus their flow is essentially dissipationless with a drift velocity $v_d = E_x / B$ in the $y$ direction. The heat current density $J_y^Q$ can be described by the law governing reversible processes, $J_y^Q = TJ_y^S$, where $J_y^S = v_d \mathcal{S}$ is the entropy current density and $\mathcal{S} = (\pi^2/3) k_B^2 T\nu$ is the electronic entropy per unit volume, with $\nu$ the density of states[26]. Crucially, for a gapless Dirac system in the EQL, the density of states approaches an energy-independent constant, $\nu = N_f eB / (2\pi^2\hbar^2 v_F)$, where $\hbar$ is the reduced Planck constant, $v_F$ is the Dirac velocity in the field direction, and $N_f$ is an integer that counts the number of Dirac points, assuming that all Dirac points are degenerate in energy. (In our system $N_f = 1$; for a Weyl system $N_f$ is given by half the number of Weyl points.) Inserting this expression for $\nu$ into the relations for $\mathcal{S}$ and $\alpha_{xy}$ gives

$$\alpha_{xy} = \frac{1}{6}\frac{T}{v_F}\frac{ek_B^2}{\hbar^2}N_f. \tag{1}$$

In other words, in the EQL the value of $\alpha_{xy} / T$ is determined only by the Dirac velocity, by the integer degeneracy factor $N_f$, and by fundamental constants of nature. In this sense one can say that $\alpha_{xy}v_F / T$ is a universal quantity in the EQL of Dirac or Weyl materials, which depends only on fundamental constants and on the integer $N_f$. Equation (1) was predicted in ref. [7], where it was derived in terms of quantum Hall-like edge states.

Notice, in particular, that Eq. (1) has no dependence on the carrier concentration. Thus, changes in the carrier concentration or even a transition from $n$-type to $p$-type conduction do not affect the value of $\alpha_{xy}$. Empirical evidence for this lack of dependence can be seen by noting that the carrier concentration and carrier sign vary strongly within the range of $T$ and $B$ corresponding to the plateau in $\alpha_{xy}$ (see Supplementary Note 2 for more discussion). The surprising independence of $\alpha_{xy}$ on the carrier concentration apparently enables the universal plateau that we observe in $\alpha_{xy}$, even though the behavior of $S_{xx}$ and $S_{xy}$ in the EQL is more complicated. Figure 3a shows that the plateau in $\alpha_{xy} / T \approx 0.01\,AK^{-2}\,m^{-1}$. We can compare this to the theoretical prediction of Eq. (1) using the previously-measured Dirac velocity

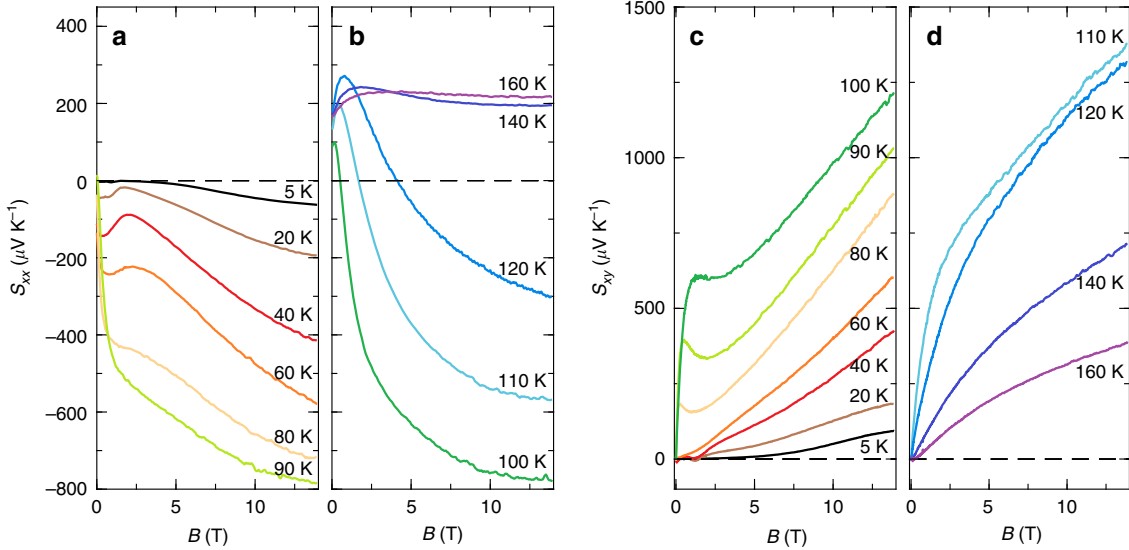

**Fig. 2 Longitudinal and transverse thermoelectric coefficients as a function of the magnetic field at different temperatures. a** $S_{xx}$ at $T \leq 90$ K, where the sign of $S_{xx}$ is negative at $B = 0$, and **b** $S_{xx}$ at $T > 90$ K, where the sign is positive at $B = 0$. **c, d** show $S_{xy}$ at $T \leq 90$ K and $T > 90$ K, respectively.

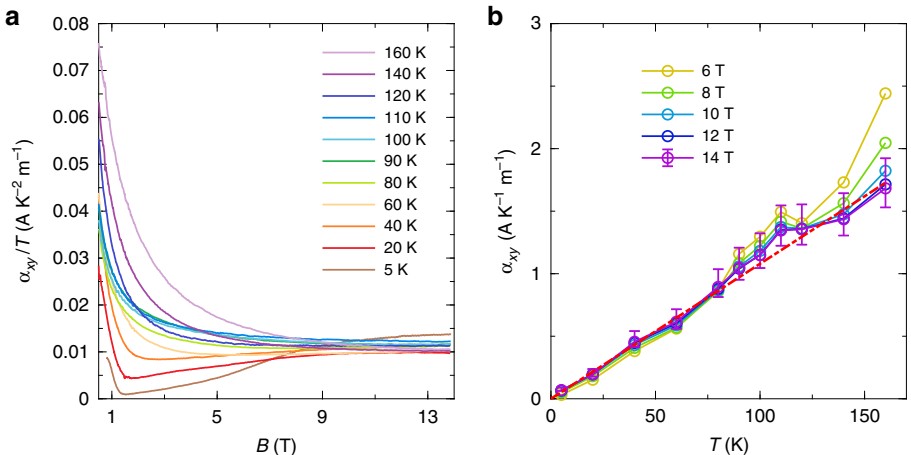

**Fig. 3 Transverse thermoelectric conductivity. a** $\alpha_{xy}/T$ as a function of $B$ at different temperatures. $\alpha_{xy}$ is independent of $B$ at high fields. **b** $\alpha_{xy}$ as a function of temperature at different fields. The red dashed line is a guide to the eye. The error bars indicate one standard error, and reflect uncertainties in the measurements of the temperature difference and sample dimensions (see Supplementary Note 1 for detail). For visual clarity, only the error bars for $B = 14$ T are shown.

$v_F \approx 3 \times 10^4$ m/s[17]. Inserting this value into Eq. (1) gives $\alpha_{xy}/T \approx 0.015$ AK$^{-2}$ m$^{-1}$, which is in excellent agreement with our measurement. (Data from another sample is presented in Supplementary Note 4). Equation (1) also predicts a lack of dependence on the disorder strength, in the sense that the value of $\alpha_{xy}$ has no dependence on the electron mobility or transport scattering time. This lack of dependence on disorder is more difficult to directly test experimentally.

It is worth emphasizing that in conventional gapped systems, such as doped semiconductors, $\alpha_{xy}$ varies with both the carrier concentration $n$ and the magnetic field $B$ in a nontrivial way[7]. In this sense the plateau in $\alpha_{xy}/T$ is a unique hallmark of three-dimensional Dirac and Weyl semimetals. (The effects of a finite band gap on $\alpha_{xy}$ are discussed in Supplementary Note 6).

Finally, we briefly comment on the possibility of making efficient thermoelectric devices using the strong magnetic field enhancement of the thermoelectric coefficients that we observe. In particular, the strong enhancement of the longitudinal and transverse thermoelectric coefficients, $S_{xx}$ and $S_{xy}$, would seem to suggest a large enhancement of the thermoelectric power

factor PF $= S_{xx}^2/\rho_{xx}$. Unfortunately, the growth in $S_{xx}$ with field is compensated by the large magnetoresistance in ZrTe$_5$ (as we discuss in Supplementary Note 2), so that no significant growth of the power factor is seen. The variation in carrier density with magnetic field $B$ also blunts the field enhancement of the thermopower, particularly at higher temperatures. Thus, direct use of ZrTe$_5$ in thermoelectric devices will likely require further research that can find a way to suppress the magnetoresistance while maintaining the large magnetothermoelectric effect.

In summary, in this article we have demonstrated a robust plateau in the thermoelectric Hall conductivity of Dirac or Weyl semimetals. This plateau is unique to three-dimensional Dirac or Weyl electrons, and gives rise to the large, field-enhanced thermoelectric response that we observe at strong magnetic field. Our findings imply that ZrTe$_5$, and three-dimensional nodal semimetals more generally, may serve as effective platforms for achieving large thermopower and other unique thermoelectric responses. Our findings also suggest a transport probe for identifying and characterizing three-dimensional Dirac materials.

## Methods

**Sample details**. Our samples are single crystals of ZrTe$_5$ grown by the tellurium flux method[12,14,17,22]. Relatively large crystals, with a typical size of $3 \times 0.4 \times 0.3$ mm, were used for transport measurements. The longest dimension is along the $a$ axis and the shortest dimension is along the $b$ axis.

**Measurements**. In our measurements, either the electrical current or the temperature gradient is applied along the $a$ axis, while the magnetic field is perpendicular to the $ac$ plane. For the thermoelectric measurements, one end of the sample is thermally anchored to the sample stage, while the other end is attached to a resistive heater (see Fig. 1b). The temperature difference between the two ends is measured by a type-E thermocouple. This difference is in the range of 100–160 mK, which is always much smaller than the sample temperature. (See Supplementary Note 1 for further details about our measurement setup).

We use a two-point method to measure the temperature difference between the hot end and cold end of the sample. (The relative advantages of two-point and four-point setups for thermoelectric measurements are discussed in Supplementary Note 1). Two Type-E thermocouples are attached to the heater stage and the heat sink, respectively (see Supplementary Fig. 2). Thermoelectric voltages are measured through gold wires attached to two ends of the sample and the voltage probes in the middle. $\rho_{xx}$ and $\rho_{xy}$ are measured by a standard four-point method.

## Data availability

The data that support the findings of this study are available from the corresponding authors upon reasonable request.

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

## Acknowledgements

Financial support for W.Z., R.B., and X.W. comes from the National Key Basic Research Program of China (No. 2016YFA0300600) and NSFC (No. 11574005, No. 11774009). Work at SUSTech was supported by Guangdong Innovative and Entrepreneurial Research Team Program (No. 2016ZT06D348), NFSC (11874193) and Shenzhen Fundamental subject research Program (JCYJ20170817110751776) and Innovation Commission of Shenzhen Municipality (Grant No. KQTD2016022619565991). Work at MIT was supported by DOE Office of Basic Energy Sciences, Division of Materials Sciences and Engineering under Award DE-SC0018945. BS was supported by the NSF STC "Center for Integrated Quantum Materials" under Cooperative Agreement No. DMR-1231319. VK was supported by the Quantum Materials program at LBNL, funded by the US Department of Energy under Contract No. DE-AC02-05CH11231. Work at Brookhaven is supported by the Office of Basic Energy Sciences, U.S. Department of Energy under Contract No. DE-SC0012704.

## Author contributions

L.F., X.W., and L.Z. conceived the idea for this study. W.Z., P.W., R.B., C.W.C., X.W., and L.Z. performed the measurements and analyzed the associated data. DY contributed to the discussion of the results. B.S., V.K., and L.F. provided theoretical analysis. R.Z., J.S., and G.G. grew the samples. B.S., V.K., X.W., and L.Z. wrote the manuscript, with input from all authors.

## Competing interests

The authors declare no competing interests.
