## [Peer Review File · Nature Communications]

Editorial Note: This manuscript has been previously reviewed at another journal that is not operating a transparent peer review scheme. This document only contains reviewer comments and rebuttal letters for versions considered at *Nature Communications* . Mentions of prior referee reports have been redacted.

Reviewers' comments:

Reviewer #1 (Remarks to the Author):

I have read the revised manuscript and the reply letter.

In the reply letter, the authors clarify most of the criticisms raised by the Referees of the first-round submission to [redacted]. The provided explanation in the reply letter is quite detailed and convincing. The main criticism in my first report (which was the term "quantized" and "universal") is now taken into account by removing these terms for the title and the text. I think this work as the first observation of a field-independent thermoelectric Hall coefficient in a Dirac material is an important discovery. This study can attract researchers who work on Dirac materials.

In this regard, I recommend this manuscript for publication in Nature Communications.

Reviewer #2 (Remarks to the Author):

In this manuscript, W. Zhang et al. report a plateau in the field dependence of the thermoelectric Hall conductivity α_{xy} in the 3D Dirac semimetal ZrTe₅. It is claimed that this plateau is independent of the field strength, disorder strength, carrier concentration, or carrier sign.

This manuscript was revised to correspond the review report for their previous submission. This review process is found to be very successful. Almost all concerns raised by all the referees are properly responded, and the scientific finding brought by this experiment is now fairly addressed.

I agree that the results are very important for further characterization of 3D Dirac materials by transport measurements, which would work as complementary to ARPES studies, as authors suggest. This will give a new perspective for understanding thermopower properties of Dirac materials. I would like to recommend the publication of this manuscript if my concerns described below are carefully addressed.

1. One of the main conclusions of this manuscript is the plateau field dependence of α_{xy}/T and, “The plateau value is independent of the field strength, disorder strength, carrier concentration, or carrier sign”. Regarding this statement, the concern about the dependence on the carrier concentration was raised in the previous review. I do not think this concern is fully resolved in this revision. This is because the experimental results verifying the statement are not well explained in this manuscript. In their reply, it is pointed out that the difference of the plateau value of α_{xy}/T in sample 1 and 2 can be explained by the difference of the Fermi velocity caused by the high sensitivity of the band structure of ZrTe₅ on the lattice deformations. This can explain the difference, but does NOT verify that the plateau is independent of the disorder strength, carrier concentration, and carrier sign.

More careful discussions for each dependence of α_{xy} should be given to improve the clarity of the manuscript. I think that the small field dependence is well shown in Fig. 3. I presume that the dependence on the carrier should be discussed by comparing the large temperature dependence of the carrier density (0 to 150 K, Fig. S6 (e)) and the linear temperature dependence α_{xy} observed in a wide temperature range (Fig. 3 (b)). On the other hand, the dependence on the disorder strength is not seemed to be carefully explained. If it is based on the difference of ρ_{xx} of sample 1 and 2, the temperature dependence of ρ_{xx} of sample 2 must be shown as Fig. 1 (c). In the current version of the manuscript, only the field dependence of ρ_{xx} and ρ_{xy} of sample 2 is shown in Supplementary Information. In

particular, the peak temperature of ρ_{xx} of sample 2 would be important for characterizing the sample dependence.

Related to this point, the magneto resistance of sample 2 is much smaller than that of sample 1. For example, $\rho_{xx} \sim 100$ m Ω ·cm at 5 K, 10 T in sample 1 (Fig. S6(a)) whereas it is only ~ 10 m Ω ·cm in sample 2 (Fig. S10 (a)). Can the authors comment on this?

2. The concern about the insufficient description for the experimental details was raised by Reviewer #2 and #3. This issue is partly resolved by the updated Supplementary Information, but it remains in a qualitative discussion. It is stated that the two-point method is preferred than the four-point method, because the thermal resistance of the sample is “high” whereas the contact thermal resistance can be made “small”. This statement must be justified by quantitative discussion. Otherwise, it is difficult for readers to assess the reliability of the measurements.

The error by the contact thermal resistance is estimated by Fig. S3. However, it is not clear how much α_{xy} (also S_{xx} and S_{xy}) is affected by this error. Since both the absolute value and the temperature dependence of α_{xy} are important results of this manuscript, the error in α_{xy} brought by δT should be quantitatively estimated.

Some typos are addressed as following,

In page 3, 7th line from the bottom of the second paragraph,

“However, this two-point method works only if the thermal resistance ...”

This part would be,

*“However, this **four**-point method works only if the thermal resistance ...”*

The term of “quantized plateau of α_{xy} ” remained in the figure caption of Fig. S12 (b) should be replaced as pointed by the previous review.

Reviewer #3 (Remarks to the Author):

In the revision, the authors indeed have supplemented important information to support their claims. However, I do have some additional concerns and would like to recommend to accept a manuscript with these concerns well-addressed.

1. Although the authors have toned down the claim of “quantized”. I am still not convinced by the claim that the thermoelectric Hall plateau is independent of disorder strength, carrier concentration or carrier sign. Strong disorders or defects in a material can have important effect on the electronic structure which could change the Fermi velocity. Thus, the Hall plateau might not be robust, as it is Fermi velocity dependent.

2. in Fig. S1, the Nernst coefficient measurement probes are not shown, which should also be labeled. Moreover, I am wondering how did the authors calculate the Nernst coefficient. Normally, the Nernst voltage ΔV is measured along the transverse direction with a length of L_y , while the temperature difference ΔT is applied along the longitudinal direction with a length of L_x . The Nernst coefficient S_{xy} is calculated according to the equation $S_{xy} = (\Delta V/L_y) / (\Delta T/L_x)$. Is the S_{xy} was also calculated in such a way? How did the authors estimate the L_x ? The two contacts with the stages are rather large, the distance between them is of high uncertainty.

3. Throughout the paper, the authors claim their crystals have very high quality and consistently behave as 3D Dirac semimetals in different studies. According to the calculation of ref. 8, Dirac semimetals at extreme quantum limit will exhibit a linear, unsaturated thermopower, which provides an immediate way to reach record-high thermopower and thermoelectric figure of merit. The power factor was predicted to have an increase with B^2 at a carrier concentration of 10^{17} cm^{-3} . In the current experimental case, ZrTe5 single crystal has even low carrier concentration of $5 \times 10^{16} \text{ cm}^{-3}$, however, it seems there is no significant growth of the power factor with magnetic fields due to the compensation of large magnetoresistance. If the theoretical prediction cannot be justified in such an ideal 3D Dirac semimetals, is it still possible to achieve an enhancement of thermoelectric figure of merit of Dirac/Weyl semimetals by applying large magnetic field? This raising question and related discussion might exceed the scope of the current study, but it is important for the readers who are interested in such a novel thermoelectric response.

4. The unit of α_{xy}/T in Page 7 paragraph 2 was wrongly given.

Response to Reviewer #1

Report: *I have read the revised manuscript and the reply letter.*

In the reply letter, the authors clarify most of the criticisms raised by the Referees of the first-round submission to [redacted]. The provided explanation in the reply letter is quite detailed and convincing. The main criticism in my first report (which was the term "quantized" and "universal") is now taken into account by removing these terms for the title and the text. I think this work as the first observation of a field-independent thermoelectric Hall coefficient in a Dirac material is an important discovery. This study can attract researchers who work on Dirac materials.

In this regard, I recommend this manuscript for publication in Nature Communications.

Reply: We thank the reviewer for this accurate summary of our work, and for recommending publication of the manuscript in its present form.

Response to Reviewer #2

Reviewer's summary: *In this manuscript, W. Zhang et al. report a plateau in the field dependence of the thermoelectric Hall conductivity α_{xy} in the 3D Dirac semimetal $ZrTe_5$. It is claimed that this plateau is independent of the field strength, disorder strength, carrier concentration, or carrier sign.*

This manuscript was revised to correspond the review report for their previous submission. This review process is found to be very successful. Almost all concerns raised by all the referees are property responded, and the scientific finding brought by this experiment is now fairly addressed.

I agree that the results are very important for further characterization of 3D Dirac materials by transport measurements, which would work as complementary to ARPES studies, as authors suggest. This will give a new perspective for understanding thermopower properties of Dirac materials. I would like to recommend the publication of this manuscript if my concerns described below are carefully addressed.

Reply: We thank the reviewer for this positive summary and for recommending publication once the reviewer's comments have been addressed. Below we have made a careful effort to fully address both of the reviewer's comments, and to correct the typos that the reviewer pointed out.

Comment 1-1: *“One of the main conclusions of this manuscript is the plateau field dependence of α_{xy}/T and, “The plateau value is independent of the field strength, disorder strength, carrier concentration, or carrier sign”. Regarding this statement, the concern about the dependence on the carrier concentration was raised in the previous review. I do not think this concern is fully resolved in this revision. This is because the experimental results verifying the statement are not well explained in this manuscript. In their reply, it is pointed out that the difference of the plateau value of α_{xy}/T in sample 1 and 2 can be explained by the difference of the Fermi velocity caused by the high sensitivity of the band structure of $ZrTe_5$ on the lattice deformations. This can explain the difference, but dos NOT verify that the plateau is independent of the disorder strength, carrier concentration, and carrier sign.*

More careful discussions for each dependence of α_{xy} should be given to improve the clarity of the manuscript. I think that the small field dependence is well shown in Fig. 3. I presume that the dependence on the carrier should be discussed by comparing the large temperature dependence of the carrier density (0 to 150 K, Fig. S6 (e)) and the linear temperature dependence α_{xy} observed in a wide temperature range (Fig. 3 (b)). On the other hand, the dependence on the disorder strength is not seemed to be carefully explained. If it is based on the difference of ρ_{xx} of sample 1 and 2, the temperature dependence of ρ_{xx} of sample 2 must be shown as Fig. 1 (c). In the current version of the manuscript, only the field dependence of ρ_{xx}

and ρ_{xy} of sample 2 is shown in Supplementary Information. In particular, the peak temperature of ρ_{xx} of sample 2 would be important for characterizing the sample dependence.”

Reply: The reviewer suggests that we provide additional discussion for each “independence” of the plateau in α_{xy} , and discuss separately the empirical evidence for the lack of dependence on magnetic field, disorder strength, carrier concentration, and carrier sign. This is a good suggestion, and in our revised submission we have attempted to provide such additional discussion.

Each of these lack of dependences is predicted by the theory in our Ref. 7, which derives the thermoelectric Hall effect in terms of quantum Hall-like edge states and agrees quantitatively with our experimental results. At an empirical level, however, the evidence for each lack of dependence can be summarized as follows:

- The lack of magnetic field dependence is evidenced by the plateau shown in Fig. 3, as indicated by the reviewer.
- The predicted plateau value of α_{xy} [Eq. (1)] has no dependence on disorder in the sense that it is independent of the electron mobility or transport scattering time. (An important caveat to this claim of disorder-independence is discussed in our response to Comment 1 by Reviewer #3, below). This lack of dependence on disorder can be shown theoretically in a straightforward way, either through the derivation using edge states in Ref. 7 or through our simpler argument in the paragraph containing Eq. (1). Nonetheless, the disorder-independence is difficult to directly confirm experimentally, since a clear demonstration would involve systematically changing the disorder strength in a single sample. Even though our results seem to agree quantitatively with the theoretical prediction, such a difficult experiment with varied disorder strength is beyond the scope of the current paper. Thus, the most conservative course of action is to make no claims as to whether this independence on disorder strength has been verified experimentally.
- The lack of dependence on carrier concentration and carrier sign is indicated by comparing Figs. 3 and S6, as the reviewer suggests. These figures show that the concentration and sign of carriers varies significantly with magnetic field, while the plateau value of α_{xy}/T remains unchanged. (One can also infer from the sign of the Seebeck coefficient in Fig. 2b that the carrier concentration is changing sign as a function of magnetic field and temperature, but that the value of α_{xy} is unaffected.)

In addition to the theoretical derivation in the paragraph containing Eq. (1), we have expanded the subsequent paragraph in order to explain the points listed above.

We have also added a comparison of the temperature-dependent resistivities for samples #1 and #2 (Fig. S11), as the reviewer suggested.

Comment 1-2: “Related to this point, the magneto resistance of sample 2 is much smaller than that of sample 1. For example, $\rho_{xx} \sim 100 \text{ m}\Omega\cdot\text{cm}$ at 5 K, 10 T in sample 1 (Fig. S6(a)) whereas it is only $\sim 10 \text{ m}\Omega\cdot\text{cm}$ in sample 2 (Fig. S10 (a)). Can the authors comment on this?”

Reply: Given the low electron concentration in our samples, the strength of the magnetoresistance at such high fields, $B \sim 10 \text{ T}$, is largely determined by the evolution of the carrier concentration with magnetic field. As discussed in more detail in our reply to Reviewer #3 comment 3, below, the electron concentration in our samples changes as a function of magnetic field and temperature due to the presence of a heavy electron band with a band edge that is close in energy to the Dirac point (see Fig. S6). Slight differences in the electron concentration of the sample can affect the magnetic field strength associated with the corresponding Lifshitz transition, leading to strong variations in the magnetoresistance from one sample to another. From our perspective, this mechanism is likely responsible for the large difference in magnetoresistance between the two samples. It is worth noting, however, that both samples show strong magnetoresistance effects that look qualitatively similar. The primary difference is only that sample #2 has a slightly higher value of the field associated with the inflection point in $\rho_{xx}(B)$, which is consistent with its slightly larger electron density at $B = 0$.

We have added three sentences to the end of the first paragraph of text in the Supplementary section IV discussing this likely origin of the difference between samples.

Comment 2: “The concern about the insufficient description for the experimental details was raised by Reviewer #2 and #3. This issue is partly resolved by the updated Supplementary Information, but it remains in a qualitative discussion. It is stated that the two-point method is preferred than the four-point method, because the thermal resistance of the sample is “high” whereas the contact thermal resistance can be made “small”. This statement must be justified by quantitative discussion. Otherwise, it is difficult for readers to assess the reliability of the measurements.

The error by the contact thermal resistance is estimated by Fig. S3. However, it is not clear how much α_{xy} (also S_{xx} and S_{xy}) is affected by this error. Since both the absolute value and the temperature dependence of α_{xy} are important results of this manuscript, the error in α_{xy} brought by δT should be quantitatively estimated.”

Reply: We thank the reviewer for this suggestion. We have added out a quantitative discussion of this issue in the Supplementary Information, comprising the long paragraph just before Fig. S3.

Comment 2: “Some typos are addressed as following,

In page 3, 7th line from the bottom of the second paragraph,

“However, this two-point method works only if the thermal resistance ...”

This part would be,

*“However, this **four**-point method works only if the thermal resistance ...”*

The term of “quantized plateau of α_{xy} ” remained in the figure caption of Fig. S12 (b) should be replaced as pointed by the previous review.”

Reply: These typos have been corrected. We thank the reviewer for pointing them out.

Response to Reviewer #3

Reviewer's summary: *“In the revision, the authors indeed have supplemented important information to support their claims. However, I do have some additional concerns and would like to recommend to accept a manuscript with these concerns well-addressed.”*

Reply: We thank the reviewer for this accurate summary of our work, and for recommending publication. In this revision we have been careful to address each of the reviewer's four comments listed below.

Comment 1: *“Although the authors have toned down the claim of “quantized”. I am still not convinced by the claim that the thermoelectric Hall plateau is independent of disorder strength, carrier concentration or carrier sign. Strong disorders or defects in a material can have important effect on the electronic structure which could change the Fermi velocity. Thus, the Hall plateau might not be robust, as it is Fermi velocity dependent.”*

Reply: This comment by Reviewer #3 echoes the Comment 1-1 by Reviewer #2. That is, the reviewer asks for increasing discussion and justification of the independence of the plateau in α_{xy} as a function of disorder, carrier concentration, and carrier sign. We therefore refer to our response to this previous comment.

The reviewer also points out that strong disorder can renormalize the electronic structure and change the Fermi velocity. This is an important caveat to our claim, and we thank reviewer for pointing it out. We have added a comment about this caveat to the end of the paragraph that concludes at the top of page 8.

Comment 2: *“in Fig. S1, the Nernst coefficient measurement probes are not shown, which should also be labeled. Moreover, I am wondering how did the authors calculate the Nernst coefficient. Normally, the Nernst voltage ΔV is measured along the transverse direction with a length of L_y , while the temperature difference ΔT is applied along the longitudinal direction with a length of L_x . The Nernst coefficient S_{xy} is calculated according to the equation $S_{xy} = (\Delta V/L_y) / (\Delta T/L_x)$. Is the S_{xy} was also calculated in such a way? How did the authors estimate the L_x ? The two contacts with the stages are rather large, the distance between them is of high uncertainty.”*

Reply: We thank the reviewer for their careful reading, and for pointing out that more details about the Nernst measurement are useful. In response, Figure S1 has been updated so that the Nernst probes are labeled. As for the calculation of the Nernst coefficient, we have indeed used the equation $S_{xy} = (\Delta V/L_y) / (\Delta T/L_x)$, as the reviewer suggests. L_x is measured as the inner distance between the heater and the heat sink

(copper block), which is also the inner distance between two silver paste pads. This can be measured to a relatively high accuracy, ± 0.008 mm. The segments under silver paste are not included in L_x . Correspondingly, ΔT is supposed to be the temperature difference across L_x . Since what was measured in our experiment is the temperature difference between the heater and the heat sink, $\Delta T'$, there is a difference between ΔT and $\Delta T'$ due to the contact thermal resistance. This error in ΔT has been carefully estimated and discussed in the Supplementary Information, as seen in Fig. S3 and the corresponding discussion. In the revision, the error bars for α_{xy} are included to reflect the error due to ΔT and measurement uncertainties in the sample dimensions.

Comment 3: *“Throughout the paper, the authors claim their crystals have very high quality and consistently behave as 3D Dirac semimetals in different studies. According to the calculation of ref. 8, Dirac semimetals at extreme quantum limit will exhibit a linear, unsaturated thermopower, which provides an immediate way to reach record-high thermopower and thermoelectric figure of merit. The power factor was predicted to have an increase with B^2 at a carrier concentration of 10^{17} cm^{-3} . In the current experimental case, ZrTe5 single crystal has even low carrier concentration of $5 \times 10^{16} \text{ cm}^{-3}$, however, it seems there is no significant growth of the power factor with magnetic fields due to the compensation of large magnetoresistance. If the theoretical prediction cannot be justified in such an ideal 3D Dirac semimetals, is it still possible to achieve an enhancement of thermoelectric figure of merit of Dirac/Weyl semimetals by applying large magnetic field? This raising question and related discussion might exceed the scope of the current study, but it is important for the readers who are interested in such a novel thermoelectric response.”*

Reply: The referee points out that our crystals do not show a nonsaturating increase in the power factor as a function of magnetic field, even though such an increase is predicted in our Ref. 8 for 3D Dirac semimetals. This is correct, and the reason for the apparent discrepancy comes down to an assumption of the theory in Ref. 8 that is not met in our samples. In particular, Ref. 8 assumes that the electron concentration n is fixed as a function of magnetic field and temperature. In our crystals, however, n evolves as a function of both parameters, as discussed on page 4 of our manuscript. This evolution has been well-characterized in previous studies (our Refs. 22-23), and is attributed to the presence of an additional heavy band, which does not contribute significantly to transport but which has a band edge close to the energy of the Dirac point. The large density of states associated with this band allows the carrier concentration to evolve with temperature and magnetic field, which has the effect of moving the chemical potential relative to the band edge. Unfortunately this shifting of the chemical potential ends up blunting the growth of the Seebeck coefficient (and even inverting its sign, as shown in Fig. 2).

In other words, the lack of a nonsaturating growth in the power factor does not indicate that our system is not a good Dirac semimetal, but only that n has a nontrivial evolution with B and T due to the presence of a nearby, trivial band. We have added some additional sentences to the penultimate paragraph of the manuscript in order to explain this issue more fully.

Comment 4: *“The unit of α_{xy}/T in Page 7 paragraph 2 was wrongly given.”*

Reply: This typo has been corrected. We thank the reviewer for pointing it out.

REVIEWERS' COMMENTS:

Reviewer #2 (Remarks to the Author):

In this revision, the manuscript is much improved by responding all concerns raised in the previous round. I find that important findings are presented with better clarity and more quantitative discussions. It should be properly appreciated that the authors address unresolved issues, such as empirical evidence for the independence against the disorder strength, with sincerity in this revision. I think this work is important for further understanding of a thermoelectric property of a Dirac material and the unresolved issues will call further experimental and theoretical works in future.

I do recommend the publication of this manuscript in Nature Communications.

Reviewer #3 (Remarks to the Author):

In the response letter, the authors have addressed my previous concerns. The observation of a thermoelectric Hall plateau in 3D Dirac semimetal would attract the attention of researchers from the topological materials field. Therefore, I would like to recommend this revised manuscript for publication in Nature Communications.